# Metallaphotocatalytic triple couplings for modular synthesis of elaborate *N*-trifluoroalkyl anilines

Ting Zhou[1], Zhong-Wei Zhang[1], Jing Nie [1], Fuk Yee Kwong [2], Jun-An Ma [1] ✉ & Chi Wai Cheung [1,2] ✉

The integration of trifluoromethyl groups and three-dimensional quaternary carbon moieties into organic molecules has emerged as a prominent strategy in medicinal chemistry to augment drug efficacy. Although trifluoromethyl (hetero)aromatic amines and derivatives are prevalent frameworks in pharmaceuticals, the development of trifluoromethyl-embedded, intricately structured alkyl amine scaffolds for medicinal research remains a significant challenge. Herein, we present a metallaphotoredox multicomponent amination strategy employing 3,3,3-trifluoropropene, nitroarenes, tertiary alkylamines, and carboxylic acids. This synthetic pathway offers notable advantages, including the accessibility and cost-effectiveness of starting materials, high levels of chemo- and regioselectivity, and modularity. Furthermore, this approach enables the synthesis of a broad spectrum of aniline compounds featuring both trifluoromethyl group and distal quaternary carbon motifs along the aliphatic chains. The accelerated access to such elaborate *N*-trifluoroalkyl anilines likely involves three sequential radical-mediated coupling events, providing insightful implications for the retrosynthesis of potential compounds in organic synthesis and drug discovery.

The strategic incorporation of fluorine atom and fluoroalkyl groups into drug molecules has emerged as a fundamental approach for fine-tuning physicochemical properties and optimizing the structure-activity relationship of pharmaceuticals[1–3]. Among various fluorination and fluoroalkylation strategies, the trifluoromethyl (CF$_3$) group is particularly favored in medicinal chemistry for its ability to augment drug effectiveness through improved solubility, lipophilicity, metabolic stability, and protein-ligand interactions[4,5]. Despite the prevalence of trifluoromethyl arenes and heterocycles in pharmaceuticals[6,7] (Fig. 1a, left), the exploration of trifluoromethyl-based aliphatic bioactive molecules[8,9] has been somewhat limited, with the 2,2,2-trifluoroethyl moiety being the most commonly incorporated group in drug design[6,7]. The transition towards employing three-dimensional *sp*$^3$-hybridized carbon structures, in place of planar *sp*$^2$-

hybridized carbons, as alternative bioisosteres, represents an innovative strategy for designing highly effective drug candidates[10–12]. This shift aims to leverage the advantageous pharmacological properties, including metabolic stability, target affinity and site specificity. Yet, the development of intricate trifluoroalkyl molecules, akin to approved drugs like efavirenz and alpelisib[6,7] which feature quaternary carbons and *α*-CF$_3$ substituents (Fig. 1a, right), remains a work in progress.

Amines, particularly anilines, play a pivotal role as structural scaffolds in the creation of biologically active compounds and the synthesis of pharmaceuticals[13–15]. Introducing a trifluoroalkyl group to the nitrogen atom in aniline drugs, instead of the traditional acyl and sulfonyl groups, offers an alternative strategy to reduce oxidation and improve bioavailability[16,17]. Furthermore, the synthesis of intricate *N*-trifluoroalkyl aniline drugs, exemplified by mapracorat[6,18] (Fig. 1b, left),

[1]Department of Chemistry, Tianjin Key Laboratory of Molecular Optoelectronic Sciences, Frontiers Science Center for Synthetic Biology (Ministry of Education), Tianjin University, Tianjin 300072, P. R. of China. [2]State Key Laboratory of Synthetic Chemistry and Department of Chemistry, The Chinese University of Hong Kong, Shatin, New Territories, Hong Kong 999077, P. R. of China. ✉e-mail: majun_an68@tju.edu.cn; cw.cheung@cuhk.edu.hk

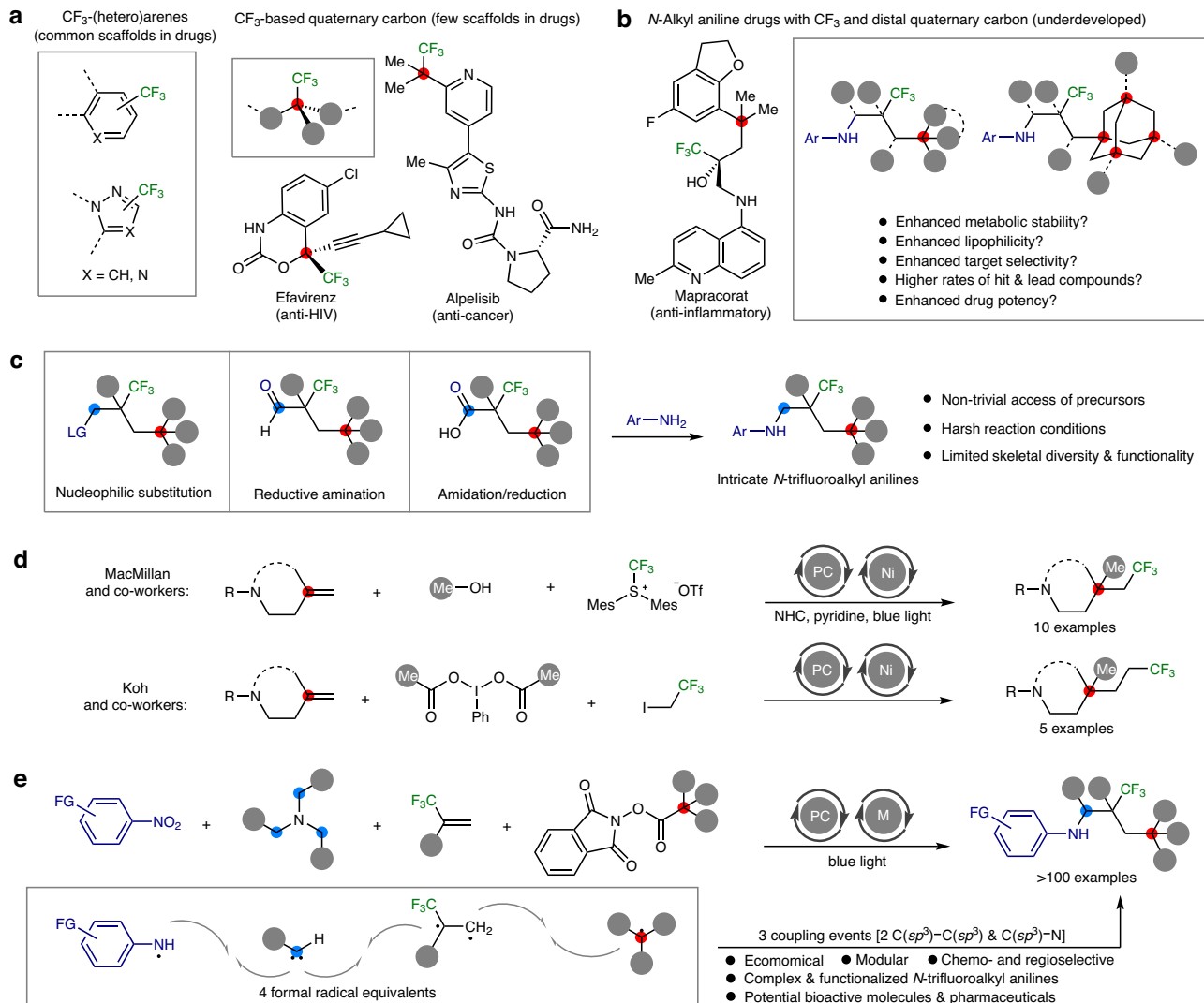

**Fig. 1 | Trifluoromethylated bioactive molecules and the construction of intricately structured complex N-trifluoroalkyl anilines. a** Trifluoromethyl (CF3) groups are widespread in pharmaceuticals, but three-dimensional CF3-based aliphatic drugs remain uncommon. **b** Intricate anilines with CF3 and remote quaternary carbons could be potentially more potent drug but remains underdeveloped. **c** Intricate N-fluoroalkyl anilines are non-trivial and challenging to access. **d** Photocatalytic multicomponent reactions have been developed to access complex CF3-substituted alkyl amines. **e** Metallaphotocatalysis would offer a modular approach to access elaborate N-trifluoroalkyl anilines using simple feedstocks. The grey circles represent hydrogen, alkyls, aryls or heteroatoms. PC, photocatalyst; M, transition metal catalyst; Me, methyl; Cbz, benzyloxycarbonyl; Ar, aryl; LG, leaving group; FG, functional groups.

incorporating both trifluoromethyl groups and distal quaternary carbon centers, illustrates a promising direction for designing potent pharmaceuticals. By expanding the molecular complexity and functionality of N-trifluoroalkyl chains, we can generate a wider array of pertinent aniline derivatives, facilitating extensive structure-activity relationship (SAR) studies and the discovery of novel drugs with enhanced physicochemical properties and drug efficacy[19,20] (Fig. 1b, right). The current methodologies for synthesizing N-trifluoroalkyl anilines primarily involve the use of preformed complex trifluoroalkyl electrophiles in various synthetic routes such as substitution, reductive amination, and amidation/reduction sequences with anilines[21–24] (Fig. 1c). However, the preparation of these substrates is often challenging and limited, which largely narrows the scope of attainable N-trifluoroalkyl anilines and derivatives.

Over the last decade, innovative approaches such as cross-coupling[25], hydroamination[26,27], C–H functionalization[28], and imine addition[29] have emerged as powerful strategies for the synthesis of a wide variety of amines and derivatives[30]. Within this realm, photocatalytic amination reactions, particularly the multicomponent version, stand out for their ability to efficiently combine simple and readily available chemical feedstocks into complex amines[31–34]. A pivotal advancement by MacMillan and co-workers was the introduction of a dual nickel/photoredox-catalyzed 1,2-dialkylation process that incorporates methanol and an electrophilic trifluoromethylating agent into amino-substituted alkenes[35] (Fig. 1d, top). Additionally, Koh and co-workers developed a method that leverages (diacetoxyiodo)benzene and 1,1,1-trifluoro-2-iodoethane for metallaphotocatalytic 1,2-dialkylation of amino-based alkenes[36] (Fig. 1d, bottom). These pioneering studies illustrate the effectiveness of photoinduced multicomponent reactions in enhancing molecular complexity and broadening the diversity of CF3-substituted alkyl amine compounds via the challenging construction of two contiguous C($sp^3$)–C($sp^3$) bonds[35–38], offering streamlined disconnections in synthetic chemistry.

Here, we introduce a metallaphotoredox-catalyzed amination process that employs nitroarenes[39], tertiary alkylamines[40,41], 3,3,3-

trifluoropropene[42], and redox-active esters derived from aliphatic carboxylic acids[43,44], inspired by these significant developments (Fig. 1e). These accessible and cost-effective reactants formally serve as precursors for aminyl radicals, carbenes, 1,2-dicarboradicals, and tertiary carbon radicals, respectively. Leveraging the synergistic effect of dual transition metal/photoredox catalysts[45], the four substrates participate in three distinct coupling events, orchestrated in a modular and systematic fashion, to synthesize an array of elaborate *N*-trifluoroalkyl anilines. Particularly, the structurally diverse nature of nitroarenes and carboxylic acids greatly enriches the complexity and functionality of the building molecules. This advancement broadens the chemical landscape, facilitating the construction of intricately structured *N*-trifluoroalkyl anilines for organic synthesis and pharmaceutical development.

## Results

### Reaction optimization

In our previous research, we successfully developed a nickel/photoredox-catalyzed three-component amination process that utilizes nitroarenes, tertiary alkylamines, and redox-active esters for synthesizing architecturally complex *N*-alkyl anilines[46]. Building upon this groundwork and employing similar reaction parameters, we aimed to test the feasibility of a more challenging four-component variation. This new approach incorporated an additional reactant, 3,3,3-trifluoropropene[42] (**F1**), known for its utility in 1,2-functionalization reactions[47–49]. The investigation commenced with the use of 4-nitroanisole (**N1**, 1.0 equiv.), 3,3,3-trifluoropropene (**F1**, 1 atm.), a redox-active ester derived from 1-adamantanecarboxylic acid (**R1**, 3.0 equiv.), and a range of tertiary alkylamines (**A1 – A6**) (Supplementary Table 1). These components served as the arylamino source, trifluoromethylated skeleton, $sp^3$-hybridized tertiary carbon precursor, and methylene linker, respectively. Employing a metallaphotoredox catalysis system consisting of [Ir(dF(CF$_3$)ppy)$_2$(dtbpy)] PF$_6$ (**PC1**, 4 mol %) and nickel(II) nitrate/bathophenanthroline (BPhen) [Ni(NO$_3$)$_2$•6H$_2$O/**L1**, 20 mol %], *N,N*-dimethylaminocyclohexane (**A1**, 6.0 equiv.) emerged as the optimal methylenating agent. The combination of dimethyl sulfoxide (DMSO) and 1,2-dimethoxyethane (DME) was identified as the ideal solvent system for the reaction. With the addition of Hantzsch ester (**HE**, 2.0 equiv.) and triflic acid (HOTf, 3.0 equiv.) serving as the reductant and proton source, respectively, a smooth reaction ensued at an ambient temperature of ~80 °C. This process yielded the intricate *N*-trifluoroalkyl aniline **1**, feathering both a CF$_3$ group and a remote adamantyl moiety attached to the *N*-aliphatic chain, in 76% yield (Fig. 2; Supplementary Table 1, entry 18). Notably, the transformation exhibited both chemo- and regioselectivity, without any detectable side-products from the two- or three-component reactions. Control

experiments further affirmed that the photocatalyst, Ni salt, and Hantzsch ester are crucial for enhancing the product yields.

### Substrate scope

Equipped with the optimal conditions, we delved into the scope of nitroarenes (**N1 – N52**) in the four-component amination process (Fig. 3). We found that nitroarenes featuring various substitution patterns, including *para-* (**N1**), *meta-* (**N2**), and *ortho-*substitutions (**N3**), as well as di- (**N37 – N42**) and tri-substitutions (**N43**), were all compatible substrates. This protocol demonstrated remarkable tolerance for a wide array of functional groups, enabling the synthesis of a diverse array of *N*-trifluoroalkyl anilines (**1 – 52**). These compounds were adorned with a rich variety of functional moieties, including esters (**4** and **21**), *tert*-alkyls (**5** and **15**), aryls (**8**, **9** and **24**), thioethers (**6**, **7**), ethers (**1 – 3**, **14**, **23**, **25**, **37**, **41**, and **43**), fluoroalkoxies (**10**, **11**, and **19**), trifluoromethyls (**20**, **27**, and **38**), chloros (**12**, **17**, **18**, **26**, and **29**), fluoros (**13**, **16**, and **42**), bromos (**39** and **40**), nitriles (**22**), protected (**36**) and unprotected aldehydes (**28**), ketones (**31**), and alkenyls (**30**). The inclusion of naphthalene (**32** and **44**) into the nitroarenes was successful as well in the transformations. Next, our exploration extended to the synthesis of *N*-trifluoroalkyl anilines featuring heteroaryl substitutions as well as N-, S-, and O-fused heterocycles, employing corresponding nitroaromatic compounds that included pyridine (**33** and **45**), thiophene (**34**), tetrahydropyran (**35**), pyrrole (**46**), benzothiophene (**47**), indole (**48**), indazole (**49**), benzodioxole (**50** and **51**), and dibenzofuran (**52**). Furthermore, the reaction was amendable to gram-scale manipulation, yielding the product **14** with similar efficiency as observed in the 0.2 mmol-scale reaction. The intricate structure of *N*-trifluoroalkyl aniline compounds was further validated through the X-ray crystallography of compound **14** (see Supplementary Information). Highly electron-deficient nitroarenes, including 4-nitrobenzotrifluoride, 4-nitrobenzonitrile, 4-nitroacetophenone, and 4-nitrobenzophenone, were not tolerated and proved non-productive under the reaction conditions. Nevertheless, this broad compatibility with a variety of functionalities and heterocycles significantly extends the versatility of *N*-trifluoroalkyl aniline derivatives, opening new avenues for the discovery of novel biologically active molecules and the advancement of pharmaceutical research.

Aliphatic carboxylic acids represent cost-effective and structurally diverse building blocks in organic chemistry. Transforming these acids into redox-active esters through the straightforward addition of a photosensitive *N*-hydroxyphthalimide group has emerged as an efficient strategy for generating $sp^3$-hybridized carbon radicals, facilitating various bond formation reactions[43,44]. Under our reaction protocol, a broad spectrum of tertiary alkyl carboxylic acid redox-active esters (**R2 – R29**) could be utilized as reactants (Fig. 4). The synthesis process enabled the creation of *N*-trifluoroalkyl anilines (**53 – 64**) with

**Fig. 2 | The optimized conditions for the metallaphotocatalytic four-component amination.** Me, methyl; Et, ethyl; Ph, phenyl; Ad, 1-adamantyl.

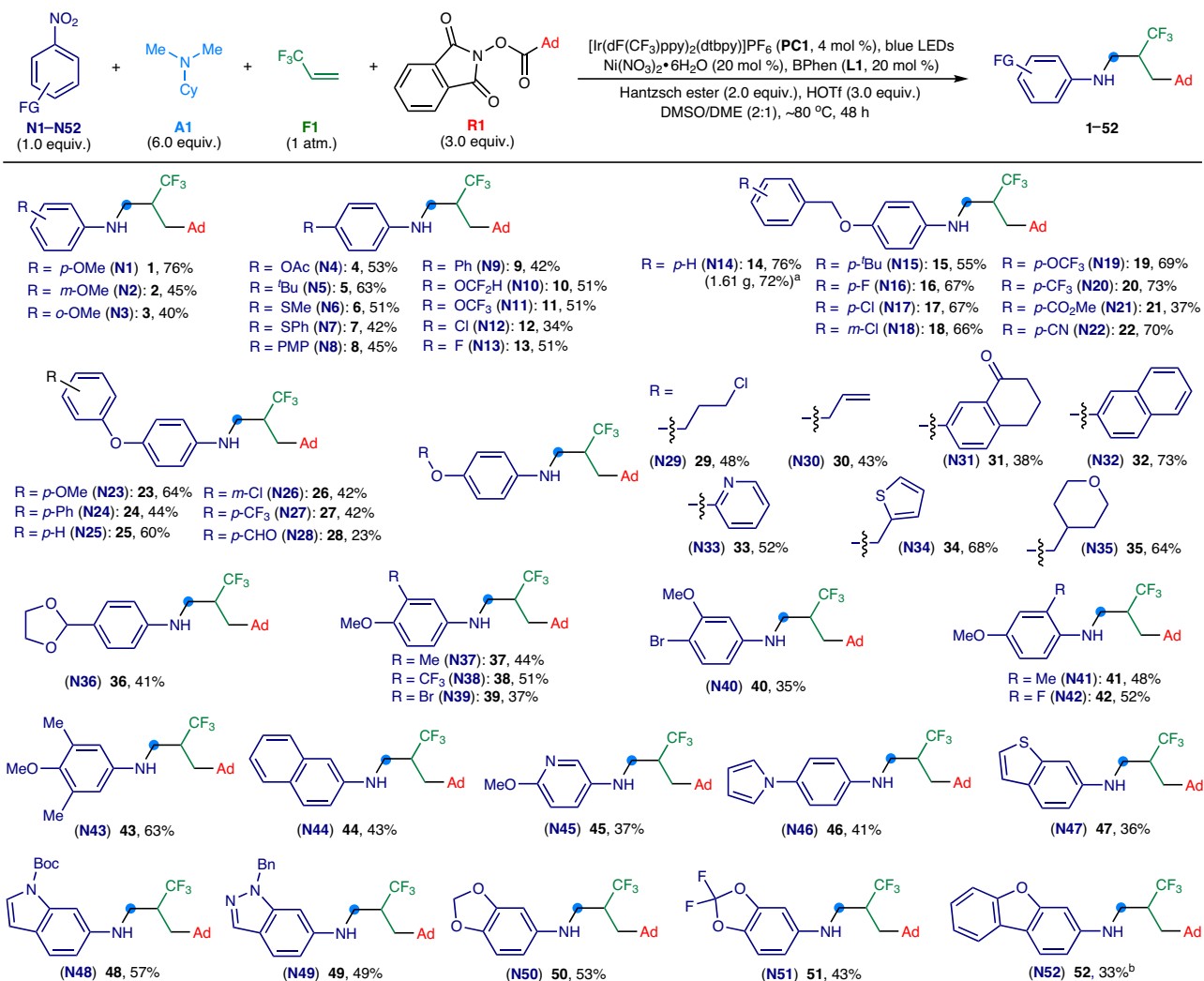

**Fig. 3 | Scope of nitroarenes.** General reaction conditions: nitroarene (**N1**–**N52**, 1.0 equiv., 0.15 mmol), tertiary alkylamine (**A1**, 6.0 equiv., 0.90 mmol), 3,3,3,-trifluoropropene (**F1**, 1 atm.), redox-active ester (**R1**, 3.0 equiv., 0.45 mmol), **PC1** (4 mol %), Ni(NO₃)₂•6H₂O (20 mol %), bathophenanthroline (BPhen, **L1**: 20 mol %), Hantzsch ester (2.0 equiv., 0.30 mmol), HOTf (3.0 equiv., 0.45 mmol), DMSO/DME (v/v = 2:1, 6 mL), -80 °C, blue LEDs (30 W, 456–460 nm), 48 h. ᵃGram-scale synthesis based on 5 mmol of **N14**, 96 h. ᵇKessil LEDs (40 W, 456 nm) were used instead. Ad, 1-adamantyl; Me, methyl; Ac, acetyl; ᵗBu, *tert*-butyl; Ph, phenyl; PMP, *p*-methoxyphenyl; Bn, benzyl; Boc, *tert*-butyloxycarbonyl.

a range of quaternary carbon-centered skeletons, including the dimethyl- (**R2**), trifluoro- (**R3**), monobromo- (**R4**) and oxo-substituted (**R5**) variants of 1-adamantane, octahydro-2,5-methanopentalene (**R6**), bicyclo[2.2.2]octane (**R7**), bicyclo[3.1.1]heptane (**R8**), bicyclo[1.1.1] pentane (**R9**), cubane (**R10**), 1-methylcyclohexane (**R11**), 4-methyltetrahydropyran (**R12**), and 2-methylpentane (**R13**). These three-dimensional aliphatic moieties can function as bioisosteres for flat aromatic rings in drug discovery[12]. Furthermore, the protocol proved compatible with secondary (**R14** – **R24**) and primary carboxylic acid redox-active esters (**R25** – **R29**) as coupling partners. A diverse array of aliphatic groups, including 2-adamantyl (**65**), isopropyl (**66**), 3-heptyl (**67**), 2-butyl (**68**), 3-pentyl (**69**), cyclopropyl (**70**), cyclopentyl (**71**), cyclohexyl (**72 and 73**), cycloheptyl (**74**), bicycloheptyl (**75**), adamandylmethyl (**76**), 2,2-dimethylpropyl (**77**), ethoxymethyl (**78**), cyclopentylmethyl (**79**), and 2-cyclohexylethyl (**80**), could be integrated into the *N*-trifluoroalkyl anilines. The use of several tertiary and secondary RAEs and all primary redox-active esters tended to result in modest product yields, due to competition from two- and three-component reactions involving nitroarenes, tertiary alkylamines, and redox-active esters. Nonetheless, the four-component synthetic approach offers a direct route to a variety of *N*-trifluoroalkyl anilines

with diverse skeletal complexity. These compounds hold promise as valuable scaffolds for the development of novel, more effective drug molecules.

Our reaction protocol successfully accommodated higher-membered tertiary alkylamines and fluoro-substituted alkenes (Fig. 5). Specifically, tripropylamine (**A7**), tripentylamine (**A8**), and tris[2-(2-methoxyethoxy)ethyl]amine (**A9**) each contributed an alkyl chain in the amination reactions. This led to the production of α-ethyl (**81**), α-*n*-butyl (**82**), and α-(2-methoxyethoxy)methyl-substituted *N*-trifluoroalkyl anilines (**83**), respectively. Additionally, fluoroalkenes, such as nonafluorohex-1-ene (**F2**), tridecafluorooct-1-ene (**F3**), and 2,3,3,3-tetrafluoroprop-1-ene (**F4**) proved to be effective reaction partners, yielding β-perfluoro-substituted (**84** and **85**) and β-fluoro-β-trifluoromethyl substituted *N*-alkyl aniline derivatives (**86**). The strategic incorporation of methyl groups into drug-like molecules has been shown to significantly enhance their ADME (Absorption, Distribution, Metabolism, and Excretion) properties and overall drug efficacy[50–52]. Leveraging triethylamine (**A10**) and a variety of tertiary *sp*³-carbon-based redox active esters as substrates (**R6, R11, R12**, and **R30 – R35**), we successfully synthesized a broader scope of *N*-trifluoroalkyl anilines (**87 – 95**). These compounds, characterized by an

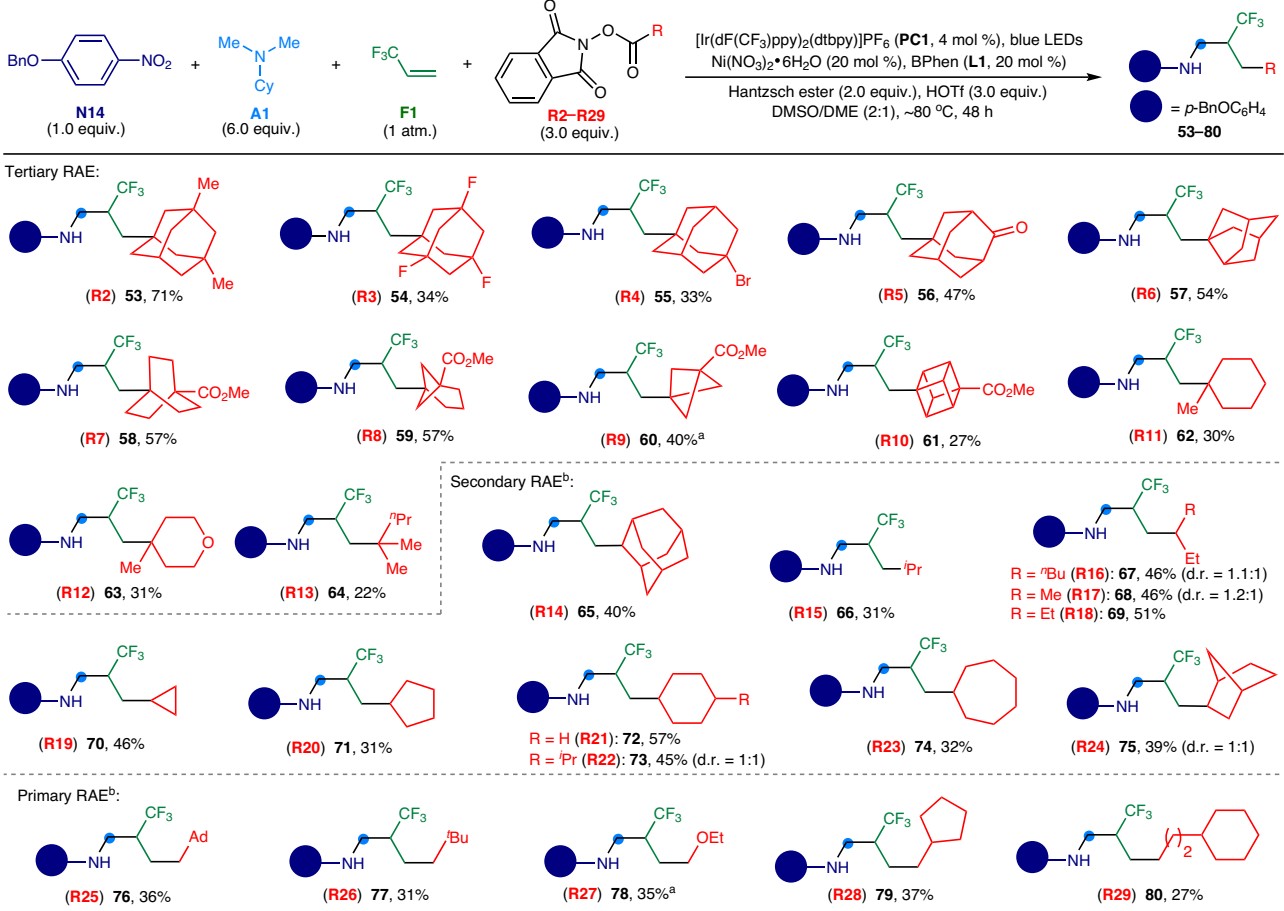

**Fig. 4 | Scope of redox active esters.** General reaction conditions: nitroarene (**N14**, 1.0 equiv., 0.15 mmol), tertiary alkylamine (**A1**, 6.0 equiv., 0.90 mmol), 3,3,3,-trifluoropropene (**F1**, 1 atm.), redox-active ester (**R2 – R29**, 3.0 equiv., 0.45 mmol), **PC1** (4 mol %), Ni(NO₃)₂•6H₂O (20 mol %), bathophenanthroline (BPhen, **L1**: 20 mol %), Hantzsch ester (2.0 equiv., 0.30 mmol), HOTf (3.0 equiv.,

0.45 mmol), DMSO/DCE (v/v = 2:1, 6 mL), -80 °C, blue LEDs (30 W, 456–460 nm), 48 h. [a]Kessil LEDs (40 W, 456 nm) were used instead. [b]Ni(bipy)Cl₂ (20 mol %) were used instead. Ad, 1-adamantyl; Me, methyl; $^{n}$Pr, $n$-propyl; $^{i}$Pr, isopropyl, $^{n}$Bu, $n$-butyl; Et, ethyl; $^{t}$Bu, $tert$-butyl; bipy = 2,2'-bipyridine.

aliphatic skeleton that includes an α-methyl group, a β-CF₃ group, and either a cyclic (**87 – 92**) or acyclic distal quaternary carbon moiety (**93 – 95**), represent a leap forward in molecular complexity. These amine compounds (**81 – 95**) would benefit from the enhanced physicochemical properties conferred by multiple fluorination, enhanced saturation, multifaceted three-dimensionality and the 'magic methyl effect'[50–52], providing potential applications in the development of more effective drug-like molecules.

Utilizing readily available and cost-efficient starting materials such as nitroarenes, tertiary alkylamines, fluoro-substituted alkenes, and carboxylic acids, our modular synthetic strategy allows for swift access to a diverse array of elaborate N-trifluoroalkyl anilines. These compounds feature varied electronic, steric and spatial properties, enabling precise structural modifications for structure-activity relationship (SAR) studies for drug discovery. This approach underscores the versatility and efficacy of our protocol, enabling the exploration of new chemical space in organic synthesis and medicinal chemistry.

## Synthetic utility

The development of structurally complex trifluoroalkyl components represents a significant challenge in medicinal chemistry, hindering the production of relevant amine compounds. A streamlined approach would involve using simpler building blocks to synthesize sophisticated trifluoroalkyl anilines, thereby facilitating the construction of novel biologically active molecules for study.

In our reaction protocol (Fig. 6a), gemfibrozil (**R36**), a lipid-regulating drug, served as a tertiary C($sp^3$)-radical precursor to synthesize the N-trifluoroalkyl aniline variants (**96** and **97**). Furthermore, nitroarene-based flutamide (**N53**), a nonsteroidal antiandrogen, and nitrofen (**N54**), an herbicide, were transformed to the trifluoroalkyl-decorated aniline derivatives (**98** and **99**).

A series of N-(4,4-dimethylpentyl) anilines (**100**) has been prepared as structural components for heterocycles **101** (Fig. 6b), aiming at synthesizing prolyl hydroxylase inhibitors **102** for treating anemia[53]. Our reaction protocol allowed for modular assembly of nitroarenes (**N12**, and **N55 – N58**), N,N-dimethylcyclohexylamine (**A1**), 3,3,3-trifluoropropene (**F1**), and redox-active esters (**R33**) derived from 2,2-dimethylpropionic acid, yielding several N-(4,4-dimethyl-2-(tri-fluoromethyl)pentyl) anilines (**103 – 107**). These CF₃-embedded analogues of **100** could potentially be modified into CF₃-substituted inhibitors **102** for biological assessment.

For the study of influenza treatments (Fig. 6c), adamantyl-incorporated N-alkyl compounds such as **108** and **109** have been synthesized[54]. Using 1-adamantylcarboxylic acid redox-active ester (**R1**), we generated several variants, N-(3-adamantyl-2-(trifluoromethyl) propyl) anilines (**110 – 113**), under our reaction protocol. The introduction of an additional CF₃ group could help develop more effective anti-influenza agents.

The N-trifluoroalkyl aniline products proved versatile for further derivatization (Fig. 6d), enhancing their functionality and structural diversity. Processes such as N-alkylation, amidation, and

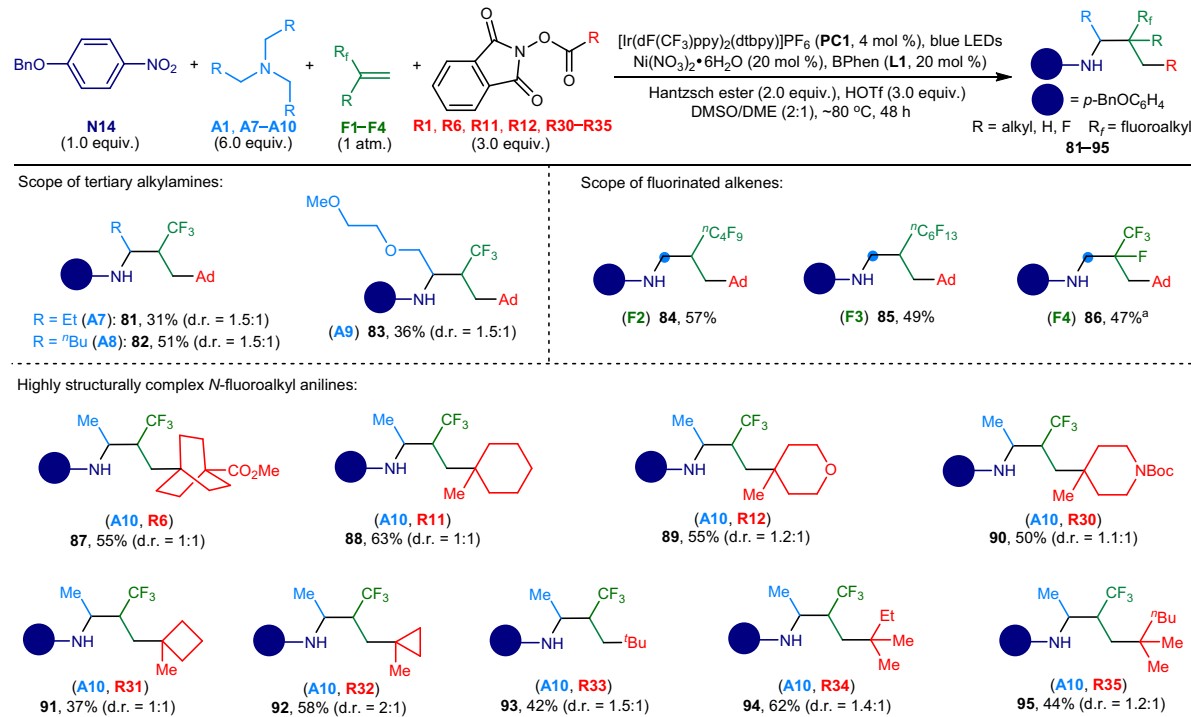

**Fig. 5 | Scope of tertiary alkylamines and fluoro-based alkenes.** General reaction conditions: nitroarene (**N14**, 1.0 equiv., 0.15 mmol), tertiary alkylamine (**A1, A7–A10**, 6.0 equiv., 0.90 mmol.), fluoro-substituted propene (**F1, F4**: 1 atm.; **F2, F3**: 20 equiv.), redox-active ester (**R1, R6, R11, R12**, and **R30 – R35**, 3.0 equiv., 0.45 mmol), **PC1** (4 mol %), Ni(NO₃)₂•6H₂O (20 mol %), bathophenanthroline (BPhen, **L1**: 20 mol %), Hantzsch ester (2.0 equiv., 0.30 mmol), HOTf (3.0 equiv., 0.45 mmol), DMSO/DME (v/v = 2:1, 6 mL), -80 °C, blue LEDs (30 W, 456–460 nm), 48 h. Unless otherwise noted, **A1, F1** and **R1** were used as reaction substrates. ᵃThe reaction was conducted at 40 °C for 24 h. Ad, 1-adamantyl; Me, methyl; Boc, *tert*-butyloxycarbonyl; ᵗBu, *tert*-butyl; Et, ethyl; ⁿBu, *n*-butyl.

sulfonamidation produced *N*-methylated (**114**) and *N*-benzylated anilines (**115**), as well as amides (**116** and **117**), formamides (**118**), and sulfonamides (**119**). Benzyloxy-substituted *N*-trifluoroalkyl aniline (**14**) underwent a tandem C – O decoupling and C – N reconstruction, delivering the *N*-benzyl-*N*-phenol isomeric variant **120**. Additionally, the oxidation of allyloxy-substituted *N*-trifluoroalkyl aniline (**30**) selectively produced the 1,2-diol derivative **121**. The Heck reaction of **30** and the Suzuki coupling reaction of the 4-chloroaniline product (**12**) also afforded the 1,3-diene (**122**) and biphenyl (**9**) derivatives. These functionalized and intricate anilines and derivatives represent valuable synthetic scaffolds for the potential development of novel drug-like molecules and more potent pharmaceuticals.

**Mechanistic study**

The synthesis of *N*-trifluoroalkyl anilines through a four-component amination process entails a series of intricate reaction sequences. To elucidate the mechanisms and identify intermediates and co-products, a comprehensive suite of experiments was undertaken:

(1) **Radical species detection** (Fig. 7a): When 2,2,6,6-tetramethylpiperidine−1-oxyl (TEMPO), acting as a radical scavenger, was introduced to the model reaction, only a minimal amount of *N*-trifluoroalkyl aniline **14** was produced. High-resolution mass spectrometry (HRMS) analysis identified several TEMPO-captured adducts (**T1 – T7**), corresponding to the adamantyl radical (**Rad-i**), the adamantyl-trifluoroalkyl radical (**Rad-ii**), the *N*-aryl-*N*-hydroxy-aminyl radical (**Rad-iii**), the *N*-aryl-aminyl radical (**Rad-iv**), the *N*-aryl-aminomethyl radical (**Rad-v**), the (α-amino) trifluoroalkyl radical (**Rad-vi**), and the α,α-di(amino)methyl radical (**Rad-vii**). Additionally, the radical clock experiment employing the RAE derived from hept-6-enoic acid yielded the ring-closing product, an *N*-trifluoroalkyl aniline featuring an incorporated cyclopentylmethyl group. This outcome suggested the formation of a hex-5-enyl radical species, which rapidly cyclizes to produce the cyclopentylmethyl radical intermediate (See Supplementary Figs. 2 and 3 for details). These findings indicated the presence of these radical intermediates, which are implicated in the reaction pathway leading to the formation of *N*-trifluoroalkyl aniline product **14**.

(2) **Identification of reaction co-products** (Fig. 7b): At the onset of the model reaction, the reaction mixture underwent HRMS analysis to identify emerging reaction intermediates and co-products (See Supplementary Fig. 4 for details). This analysis revealed a variety of species, including: (i) trifluoroalkyl amine **S1**, believed to originate from the coupling of the *N*-cyclohexyl-*N*-methylaminomethyl radical (**Rad-viii**) with **Rad-ii**; (ii) aminal **S2**, likely produced through the reaction between nitroarene **N14** and **Rad-viii**[46]; (iii) *N*-methylcyclohexylamine (**S3**), which appears to result from the nucleophilic displacement of aminal **S2** by low-valent nickel species[55], coinciding with the formation of *N*-aryl-aminomethyl-Ni species (**S4**); (iv) *N,N'*-diaryl ethylenediamine (**S5**), suggested as the dimerization outcome of the *N*-aryl-aminomethyl radical (**Rad-v**)[56,57], itself derived from the Ni−C homolysis[58–60] of **S4**; (v) trifluoroalkyl aminal (**S6**), presumed to be synthesized through the photocatalytic oxidation of aminal **S2** into α,α-diaminomethyl radical **Rad-vii**[61], followed by its integration with **Rad-ii**, and (vi) trifluoroalkyl imine (**S7**), expected to develop from the removal of amine **S3** from aminal **S6**[56]. Our hypothesis posits that **S7** further undergoes a proton-coupled electron transfer[29] (PCET) facilitated by iridium photocatalysis, forming the (α-amino)trifluoroalkyl radical species (**Rad-vi**). This radical is then believed to undergo a formal hydrogen atom transfer (HAT), leading to the formation of the *N*-trifluoroalkyl aniline product **14**.

(3) **Nitrogen intermediate exploration** (Fig. 7c): Within the photocatalytic process, nitrobenzene (**N55**) could be reduced to a variety of nitrogen-containing intermediates, including

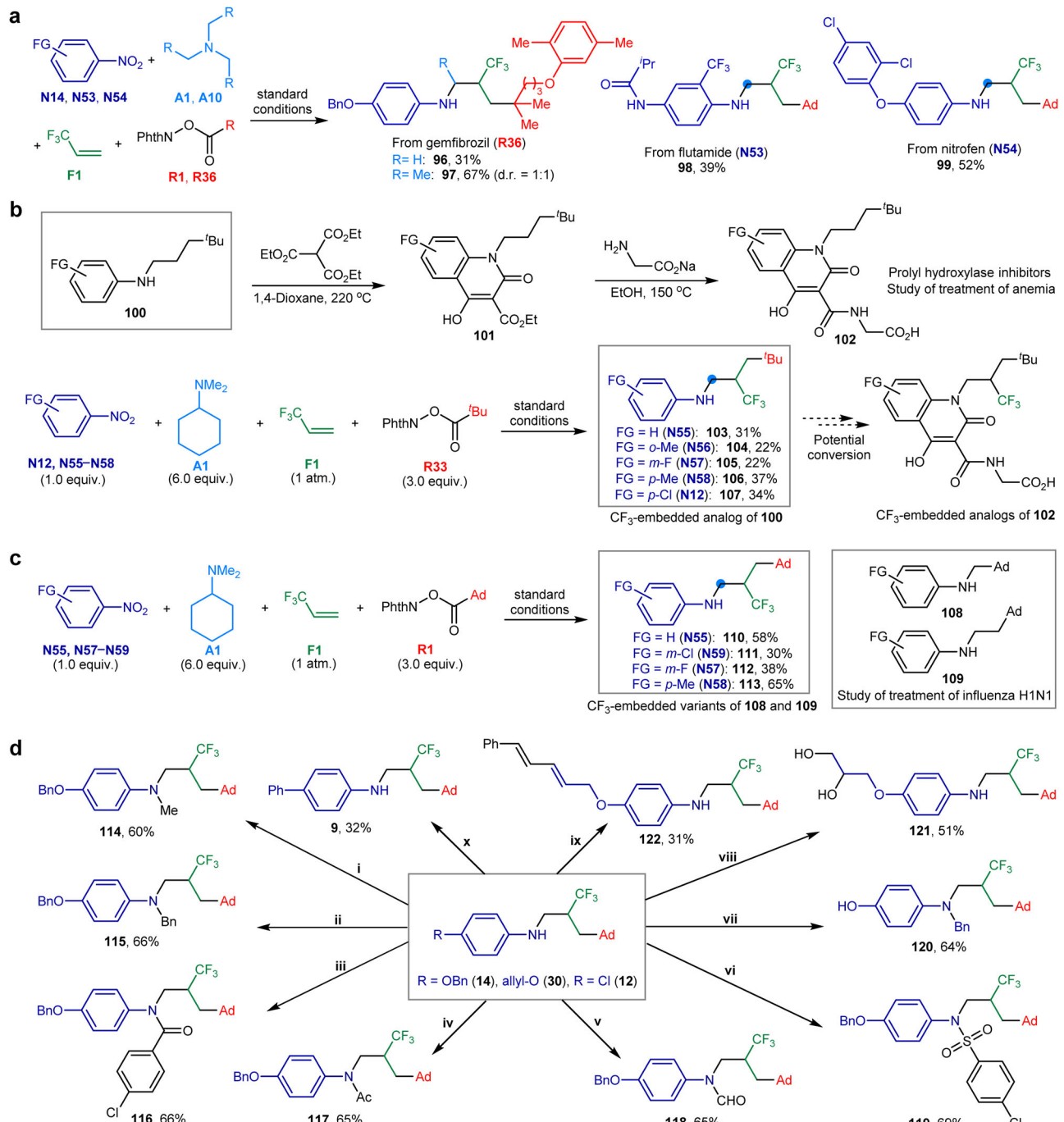

**Fig. 6 | Synthetic utility of the metallaphotocatalytic four-component amination reaction. a** Synthesis of *N*-trifluoroalkyl anilines bearing drug- and herbicide moieties. **b** Synthesis of *N*-(4,4-dimethyl-2-(trifluoromethyl)pentyl) anilines as potential prolyl hydroxylase inhibitor intermediates. **c** Synthesis of *N*-(3-adamantyl-2-(trifluoromethyl)propyl) anilines as potential anti-influenza agents. **d** Derivatization of *N*-trifluoroalkyl aniline via *N*- and peripheral functionalization: **(i)** **14**, MeI, NaH; **(ii)** **14**, BnBr, NaH; **(iii)** **14**, 4-chlorobenzoyl chloride, Et₃N; **(iv)** **14**,

acetic anhydride; **(v)** **14**, acetic anhydride, formic acid; **(vi)** **14**, 4-chlorophenylsulfonyl chloride, Et₃N; **(vii)** **14**, BBr₃, then K₂CO₃; **(viii)** **30**, potassium osmate, *N*-methylmorpholine *N*-oxide; **(ix)** **30**, (*E*)−1-bromo-2-phenylethene, Pd catalyst, Cs₂CO₃; **(x)** **12**, phenylboronic acid, Pd catalyst, K₃PO₄. Ad, 1-adamantyl; Me, methyl; Et; ethyl; *i*Pr, isopropyl; *t*Bu, *tert*-butyl; Bn = benzyl; Ac = acetyl; NPhth: phthalimidyl; FG, functional group.

nitrosobenzene (**N55-i**), *N*-phenyl hydroxylamine (**N55-ii**), azoxybenzene (**N55-iii**), azobenzene (**N55-iv**), *N,N*′-diphenyl hydrazine (**N55-v**), and aniline (**N55-vi**)[62]. Specifically, nitrobenzene was converted into *N*-trifluoroalkyl aniline (**110**) with a 58% yield under standard reaction conditions. Control experiments using **N55-i** and **N55-ii** yielded the target compound in 32% and 34% yields, respectively, while other nitrogen-containing intermediates (**N55-iii** − **N55-vi**) failed to produce the desired outcome. These results suggested that nitrosoarenes and *N*-aryl hydroxylamines could be the principal intermediates for synthesizing the target products. Experimental and analytical data led us to theorize that nitrobenzene undergoes a photocatalytic reduction, facilitated by HOTf as a proton source[46], to form **N55-i** and **N55-ii**. These intermediates are then further reduced to generate nitrogen-

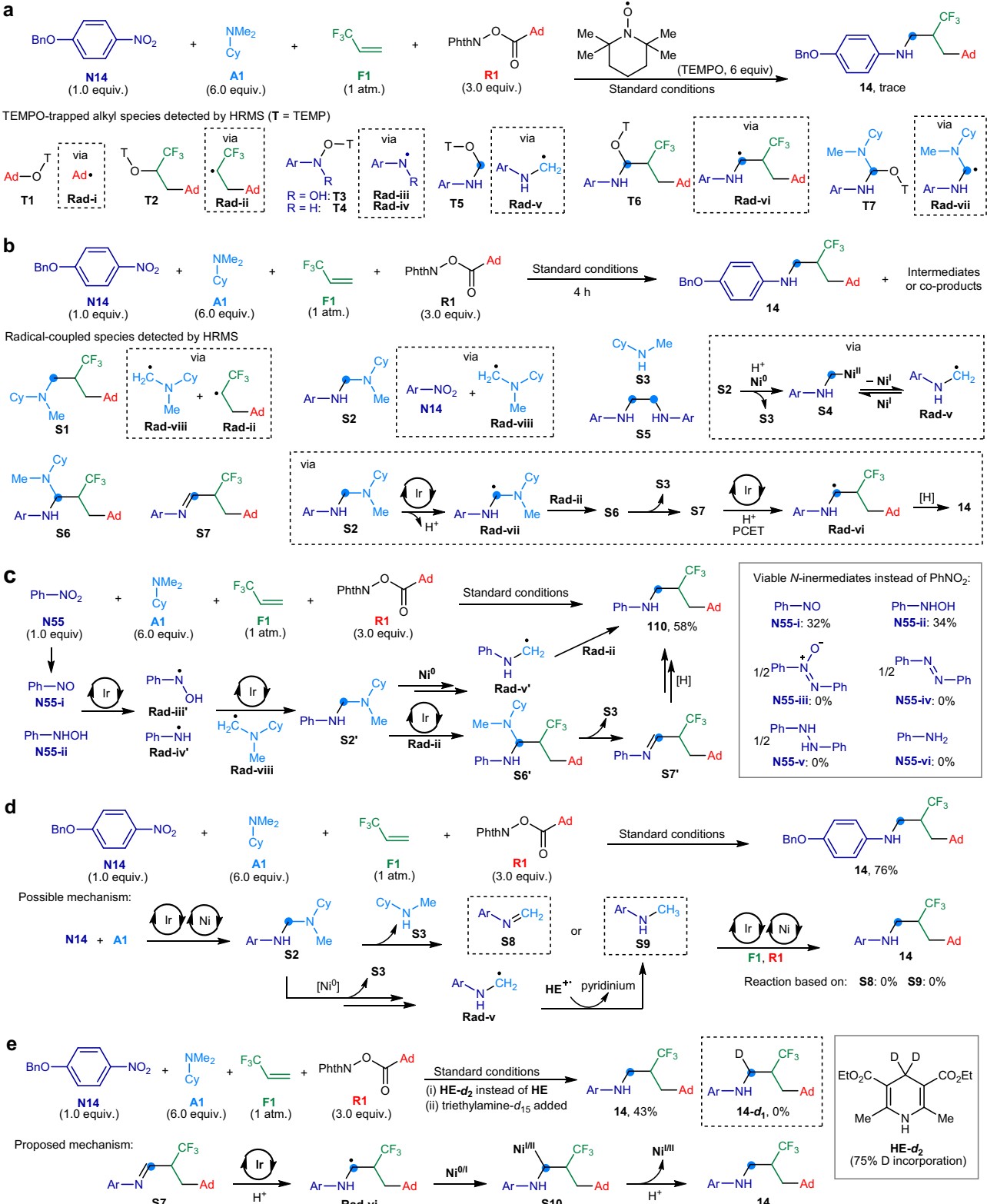

**Fig. 7 | Mechanistic study of the metallaphotocatalytic four-component amination. a** Identification of radical species using TEMPO as the radical trap. **b** Analysis of reaction co-products in the early reaction stage. **c** Examination of nitrogen-containing intermediates from nitroarenes in product formation. **d** Study of *N*-aryl imine and *N*-methyl aniline as reaction intermediates for product formation. **e** Investigation of the α-hydrogen source in *N*-trifluoroalkyl aniline product. Ar denotes the 4-benzyloxyphenyl group. Ad, 1-adamantyl; Me, methyl; Et = ethyl; Bn, benzyl; Cy: cyclohexyl; Ph, phenyl; **T** (TEMP), 2,2,6,6-tetramethylpiperidyl.

centered radical species **Rad-iii'** and **Rad-iv'**[63], which subsequently interact with **Rad-viii** to produce the aminal intermediate **S2'**[46]. This intermediate undergoes deamination[55], catalyzed by Ni, to form the *N*-phenyl-aminomethyl radical[57] (**Rad-v'**), which combines with **Rad-ii** to yield the product. Alternatively, **S2'** reacts under photocatalysis with **Rad-ii** to produce a trifluoroalkyl aminal (**S6'**). This aminal species then transitions through the formation of imine **S7'**[56] and reduction, culminating in the formation of the product.

(4) **Analysis of intermediates from nitroarenes and tertiary alkylamines** (Fig. 7d). The aminal species **S2**, arising from the photocatalytic formation of C–N bonds between nitroarene **N14** and *N,N*-dimethylcyclohexylamine **A1**, is anticipated to undergo deamination[46,56], yielding *N*-aryl imine **S8**. Concurrently, the *N*-aryl-aminoalkyl radical **Rad-v**, generated from **S2**, is expected to produce *N*-methyl aniline **S9** through HAT with Hantzsch ester. However, the reaction with in-situ formed **S8** and commercially available **S9** did not lead to the synthesis of the target compound **14**, indicating that these species are probably not the key reaction intermediates. This observation led us to conclude that **S2** and its subsequent radical form **Rad-v** are most likely the reaction intermediates, facilitating the formation of the desired product through radical-driven processes.

(5) **(α-Amino)trifluoroalkyl radical reduction study** (Fig. 7e): The α-amino-trifluoroalkyl radical species (**Rad-vi**) would engage in HAT with Hantzsch ester (**HE**) or its radical cation form (**HE⁺•**), identified as having the weakest C–H bonds {BDE (C₄–H) of **HE** = 69 kcal mol⁻¹; BDE (C₄–H) of **HE⁺•** = 31 kcal mol⁻¹}[64], resulting in the formation of *N*-trifluoroalkyl aniline **14**. However, when the model reaction employed C₄-deuterium-enriched Hantzsch ester (**HE-*d*₂**), it yielded the conventional product **14** without incorporating deuterium (Fig. 4e (i)). In addition, the C–H bond of tertiary alkylamines could serve as another HAT reagent for **Rad-vi**. However, when excess triethylamine-*d*₁₅ was added to the model reaction, no deuterium incorporation into product **14** was observed (Fig. 4e (ii)). The results implied that **Rad-vi** likely undergoes an inner-sphere electron transfer with low-valent nickel to form a Ni-(α-amino)trifluoroalkyl complex (**S10**). This complex is then protonated, releasing the target product **14**.

(6) **Stern–Volmer Quenching Analysis** (Supplementary Figs. 11–16). To elucidate the photoexcitation processes initiated by the iridium photocatalyst (**PC1, Ir^III**), we conducted Stern–Volmer quenching experiments. These experiments revealed that various quenchable species, including Hantzsch ester (**HE**), *N,N*-dimethylcyclohexylamine (**A1**), 4-benzyloxy-1-nitroarene (**N14**), redox-active ester (**R1**), and the in-situ formed nickel(II)/bathophenanthroline complex [Ni(NO₃)₂/**L1**], are capable of quenching the photoexcited state of **Ir^III***. Notably, the nitroarene (**N14**) emerged as the most potent quencher among them. This observation indicated that the photoexcited **Ir^III*** species preferentially reduce nitroarenes, leading to the formation of an **Ir^IV** species and the corresponding reduced intermediates of nitroarenes. The half-wave potential for the reduction of **N14** was determined to be −0.88 V vs SCE {E₁/₂^red [*p*-BnOC₆H₄NO₂/*p*-BnOC₆H₄NO₂⁻] = −0.88 V vs SCE in MeCN} (Supplementary Fig. 10), which falls within the redox window of the iridium photocatalyst[31], further supporting the viability of this preferential redox process. This sequence of events triggers the subsequent photocatalytic reactions.

These experiments collectively advance our understanding of the sophisticated mechanisms underpinning the synthesis of *N*-trifluoroalkyl anilines, shedding light on the intricacies of catalysts, radical formation, and reaction intermediates in achieving selective formation of product.

## Proposed mechanism

Based on the gathered experimental and instrumental evidence, we formulated a mechanism for the metallaphotocatalytic four-component amination process (Fig. 8). The iridium photocatalyst (**PC1, Ir^III**) is activated under blue light to reach a high-energy, long-lived excited state (**Ir^III***){E₁/₂^red [**Ir^IV/Ir^III**] = −0.89 V vs saturated calomel electrode (SCE)}[31]. This excited state can feasibly reduce nitroarene (**N14**) to radical anion species (**N14'**) {E₁/₂^red [*p*-BnOC₆H₄NO₂/ *p*-BnOC₆H₄NO₂⁻] = −0.88 vs SCE} (Supplementary Fig. 10)[65], especially under conditions facilitated by HOTf through oxygen protonation[46]. Further photocatalytic reduction yields nitrosoarene (**N14-i**) and *N*-aryl hydroxylamine (**N14-ii**), followed by *N*-aryl-*N*-hydroxy-aminyl radical (**Rad-iii**) and *N*-aryl-aminyl radical **Rad-iv**), respectively. Concurrently, **Ir^III*** reduces redox active esters (**RAE**) to generate alkyl radicals (**Rad-i'**), despite a redox potential of approximately −1.2 V {E₁/₂^red [R11/R11⁻] = −1.26 V vs SCE}[66]. The acidic reaction medium is believed to aid in overcoming the potential discrepancy, promoting the photoreduction of RAEs. The resulting oxidized iridium species (**Ir^IV**) {E₁/₂^red [**Ir^IV/Ir^III**] = +1.69 V vs SCE)}[31] then oxidizes *N,N*-dimethylcyclohexylamine (**A1**) to a nitrogen-centered radical cation (**A1'**) {E₁/₂^red [^iPrMe₂N⁺•/^iPrMe₂N] = +0.72 V vs SCE}[67]. This step is followed by a facile deprotonation to produce the *N*-cyclohexyl-*N*-methylaminomethyl radical (**Rad-viii**). Additionally, **Ir^IV** is regenerated to its **Ir^III** state by interacting with Hantzsch ester (**HE**) {E₁/₂^red [**HE⁺/HE**] = +0.89 V vs SCE}[68], thus completing the catalytic cycle and facilitating continuous reaction progression.

Furthermore, the [Ni^II(BPhen)](NO₃)₂ catalyst (**Ni^II**) is reduced by the excited iridium species **Ir^III***, resulting in the formation of low-valent nickel species, [Ni^I(BPhen)](NO₃) (**Ni^I**) and [Ni⁰(BPhen)] (**Ni⁰**). Given the higher dissociation tendency of the nitrate ion and the increased electrophilicity of the [Ni^II(BPhen)](NO₃)₂ complex compared to its chloride counterpart {E₁/₂^red [Ni^II(BPhen)Cl₂/Ni⁰(BPhen)] ~ −1.2 V vs SCE}[69], we hypothesized that the nitrate complex exhibits greater oxidizing properties[70]. This makes the reduction of **Ni^II** to **Ni^I** and **Ni⁰** by **Ir^III*** thermodynamically favorable.

Alkyl radicals (**Rad-i'**), generated from RAEs, react with 3,3,3-trifluoropropene (**F1**), leading to the formation of trifluoroalkyl radicals (**Rad-ii'**). Concurrently, **Rad-viii** engages with the resulting nitrogen-centered radicals (**Rad-iii** and **Rad-iv**) in a photoreduction process, yielding the *N*-aryl aminal species (**S2**). **S2** likely interacts with low-valent Ni species (**Ni⁰**) via nucleophilic substitution or oxidative addition[55], resulting in the formation of a nickel(II) *N*-aryl-aminomethyl species (**S4**)[58–60]. **S4** then undergoes Ni–C homolysis[58–60], releasing the *N*-aryl-aminomethyl radical[57] (**Rad-v**). The electrophilic trifluoroalkyl radicals (**Rad-ii'**) and nucleophilic *N*-aryl-aminomethyl radical (**Rad-v**) efficiently couple to produce the *N*-trifluoroalkyl aniline products (**123**). The polarity match of these radical species underscores the chemoselective multiple-bond connections that lead to product formation.

In an alternative pathway, **S2** is subject to photooxidation by **Ir^IV** {E₁/₂^red [((Me₂N)₂CH₂)⁺•/(Me₂N)₂CH₂] = +0.87 V vs SCE}[71], generating the α,α-di(amino)methyl radical (**Rad-vii**). The highly nucleophilic **Rad-vii** then combines with **Rad-ii'** to form the trifluoroalkyl aminal species (**S6'**). **S6'** readily dissociates into trifluoroalkyl imine (**S7'**)[56], which, upon undergoing PCET[29], produces the (α-amino)trifluoroalkyl radical species (**Rad-vi'**) {E₁/₂^red [**S7'/Rad-vi'**] ~ −1 V vs SCE}[29]. Both **Ni⁰** and **Ni^I** facilitate the inner-sphere reduction of **Rad-vi'**[58–60], followed by protonation to yield the product.

Based on the HRMS analysis (Fig. 7a and b), the relative concentration of the trifluoroalkyl radical (**Rad-ii**) -based species **T2** was significantly higher than that of the other detected radical-based species (**T3** to **T7** originated from **Rad-iii** to **Rad-vii**). This suggested that the concentrations of these other radical components are generally low. We proposed that Ni-mediated C–N and C–C bond formation reactions involving these radicals play a crucial role in the overall

**Fig. 8 | Proposed mechanism of the metallaphotocatalytic four-component amination.** Ar denotes the 4-benzyloxyphenyl group. Me, methyl; Et, ethyl; Cy, cyclohexyl; R, alkyl; NPhth, phthalimidyl.

process, beyond the free radical interactions that contribute to product formation.

We proposed that the electron-rich, low-valent nickel species, $Ni^I$ and $Ni^{II}$, capture the alkyl radical (**Rad-i′**) and trifluoroalkyl radical (**Rad-ii′**) to form Ni-alkyl complexes (**S11** and **S12**) through equilibrium processes[58–60]. This interaction is key to minimizing the undesired alkyl-alkyl dimerization and two/three-component coupling reactions. Indeed, the yields of target products dropped significantly in the reactions of primary and secondary alkyl RAEs without the Ni catalyst (Supplementary Table 2), underscoring the crucial role of Ni in stabilizing the less nucleophilic and smaller-sized primary and secondary alkyl radicals[46].

Moreover, the light/dark experiment demonstrated that the reaction occurs exclusively during the irradiation period (Supplementary Fig. 8), suggesting that it likely follows a non-chain photocatalytic mechanism rather than a radical chain process. Further $^{19}F$ NMR spectroscopic analysis suggested that fluoride is likely formed in the reaction (Supplementary Fig. 9). We surmised that the overreduction of the trifluoroalkyl radical (**Rad-ii**) via photocatalysis or Ni catalysis is inevitable, leading to the generation of a trifluoroalkyl anion that results in fluorine elimination[72,73]. However, the addition of excess RAEs and **F1** would ensure sufficient loading of **Rad-ii**, thereby compensating for the defluorination side-reaction and maintaining the reaction productivity.

This comprehensive mechanism highlights the synchronized interplay between various radicals and catalysts, leading to the efficient and selective synthesis of *N*-trifluoroalkyl aniline compounds.

In summary, we have successfully developed a metallaphotoredox-catalyzed multicomponent amination reaction that employs nitroarenes, tertiary alkylamines, 3,3,3-trifluoropropene, and carboxylic acids. This approach enables the synthesis of a diverse array of intricate, three-dimensional *N*-trifluoroalkyl aniline compounds in a modular and cost-effective fashion. Our method facilitates the efficient generation of trifluoromethylated synthetic intermediates

and compounds with pharmaceutical potential, aiming at the discovery of more potent therapeutic agents. The capability for various derivatizations enhances both the structural complexity and functional diversity of the produced *N*-trifluoroalkyl anilines. We anticipate that this versatile four-component reaction will open new avenues in the exploration of uncharted fluorinated, high $C(sp^3)$-contented structural realms, leading to the identification of novel bioactive molecules and the advancement of drug development.

## Methods
### General procedure for photocatalytic four-component amination using nitroarenes, tertiary alkylamines, 3,3,3-trifluoropropene and redox-active esters

An oven-dried, transparent 20 mL Schlenk tube equipped with a stir bar was sequentially charged with nitroarene (**N1**–**N59**, 1.0 equiv., 0.15 mmol), redox active ester (**R1**–**R36**, 3.0 equiv., 0.45 mmol), Ir[dF(CF$_3$)ppy]$_2$(dtbbpy)PF$_6$ (**PC1**, 4 mol %, 0.0060 mmol), Ni(NO$_3$)$_2$·6H$_2$O (20 mol %, 0.030 mmol), bathophenanthroline (BPhen, **L1**, 20 mol %, 0.030 mmol), and Hantzsch ester (diethyl 1,4-dihydro-2,6-dimethyl-3,5-pyridinedicarboxylate, **HE**, 2.0 equiv., 0.30 mmol). Dried dimethyl sulfoxide (DMSO, 4.0 mL) and dried 1,2-dimethoxyethane (DME, 2.0 mL) were then transferred into the tube via syringe. Subsequently, tertiary alkylamine (**A1**–**A5**, 6.0 equiv., 0.90 mmol) and triflic acid (HOTf, 3.0 equiv., 0.45 mmol) were transferred into the tube via syringe. The resulting mixture was degassed via blowing for 2 min with a needle equipped with a 3,3,3-trifluoropropene-filled-balloon (**F1**, ~1 L), after which time the tube was quickly capped with a Teflon screw cap such that it was filled with **F1** in atmospheric pressure. The reaction mixture was vigorously stirred and irradiated using 30 W blue LEDs ($\lambda$ = 456–460 nm) for 48 h under the ambient temperature of approximately ~80 °C (without the use of fans for cooling). At this point, the reaction mixture was diluted with ethyl acetate (~100 mL) and washed with water (~50 mL × 4). The organic fraction was further dried with anhydrous Na$_2$SO$_4$ and concentrated *in vacuo* with the aid

of rotary evaporator. The residue was purified by preparative thin-layer chromatography using a mixture of petroleum ether and ethyl acetate as an eluent to afford the *N*-trifluoroalkyl aniline product.

## Data availability

The experimental and analytical procedures and full spectral data are available in the supplementary materials. Crystallographic data for the structure reported in this Article has been deposited at the Cambridge Crystallographic Data Centre, under deposition numbers CCDC 2350147 (**14**). Copies of the data can be obtained free of charge via https://www.ccdc.cam.ac.uk/structures/. Additional data is available upon request from the corresponding authors.

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

## Acknowledgements

We acknowledge the National Natural Science Foundation of China [Nos. 21971186 (C.W.C.), 22271216 (C.W.C.), and 21961142015 (J.-A.M.)], the National Key Research and Development Program of China [No. 2019YFA0905100 (J.-A.M. and C.W.C.)], and the Graduate Outstanding Innovation Award Program for Humanities and Sciences 2023 Year Project from the Graduate School of Tianjin University [No. B1-2023-002 (C.W.C.)] for financial support. We also thank Zhe Feng for assistance with spectroscopic analysis and Professor Libing Zhang's group (Tianjin University) for their help with the cyclic voltammetry study.

## Author contributions

T.Z., Z.-W.Z., J.-A.M., and C.W.C. discovered the reactions. T.Z. optimized the reactions and investigated the reaction scope, synthetic utility, and mechanisms. C.W.C. wrote the manuscript with input and suggestions from T.Z., Z.-W.Z., J.N., F.Y.K., and J.-A.M. C.W.C. and J.-A.M. conceived the project, directed the research, and designed the experiments.

## Competing interests

The authors declare no competing interests.
