## [Transparent Peer Review file · Nature Communications]

Metallaphotocatalytic triple couplings for modular synthesis of elaborate N-trifluoroalkyl anilines

Corresponding Author: Professor Chi Wai Cheung

Version 0:

Reviewer comments:

Reviewer #1

(Remarks to the Author)

Cheung, Ma and coworkers report a method for the synthesis of a diverse array of three-dimensional N-trifluoroalkyl aniline compounds via a metallaphotoredox-catalyzed multicomponent amination reaction that employs nitroarenes, tertiary alkylamines, 3,3,3-trifluoropropene, and carboxylic acids as the reaction partners. The substrate scope is broad. The capability for various derivatizations enhances both the structural complexity and functional diversity of the produced N-trifluoroalkyl anilines. In general, I like this chemistry, and I would like to suggest this work to be published in *Nat. Commun.* after addressing the following concerns:

- 1) Recent synthetic application of trifluoromethyl alkenes should be cited: a) *Chem Catal.* 2022, 2, 1380-1393; b) *Angew. Chem. Int. Ed.* 2022, 61, e202212201; c) *Angew. Chem. Int. Ed.* 2023, 62, e202303218.
- 2) How about the reaction of 1,2-disubstituted trifluoromethyl alkenes instead of 3,3,3-trifluoropropene?
- 3) Radical clock experiment is suggested to be performed to support the proposed radicals.
- 4) "Macmillin and co-workers" should be "MacMillan and co-workers".
- 5) Supplementary Information: compound N31, 10, 29, 41, 42, 99, 107, 1H NMR quantity is incorrect, please check carefully.
- 6) Supplementary Information: compound R3, R24, 36 "HRMS (ESI) m/z: [M+H]⁺" should be "HRMS (ESI) m/z: [M+Na]⁺".
- 7) Supplementary Information: compound 95, "HRMS (ESI) m/z: [M+H]⁺ Calcd for C₂₅H₂₅F₃NO⁺" should be "HRMS (ESI) m/z: [M+H]⁺ Calcd for C₂₅H₃₅F₃NO⁺".
- 8) Supplementary Information. There are obvious impurities in compound 80 and 111 spectra.

Reviewer #2

(Remarks to the Author)

This manuscript details a metallaphotocatalytic triple couplings for modular synthesis of elaborate N-trifluoroalkyl anilines. The yields of this transformation are generally good and the substrate scope is broad. Although this strategy has been applied to a three-component tandem reaction (*Nat. Synth.* 2023, 2, 1171), the authors achieved a more complex coupling reaction in this work, effectively avoiding the production of side-products from the two- or three-component reactions, which is still an improvement. Series of late-stage functionalization have been demonstrated, which are impressive and suggest that this process will likely be useful to the synthetic and medicinal chemistry community. In the mechanism study, a comprehensive suite of experiments was undertaken, which revealed the reaction intermediates and provided some evidence for the possible mechanism of these metallaphotocatalytic four-component amination process. However, due to the complexity of the reaction system, the study of the mechanism may need to be more detailed. Unquestionably, this is an important advance in the field of exploration of uncharted fluorinated, high C(sp³)-contented structural realms. Overall, I think this work will be of broad interest among the synthetic organic chemistry community, and meet the bar for publication on nature communication if the authors address the issues below.

1. In line 47, the author writes: "with the trifluoroethyl moiety being the most commonly incorporated group in drug design", which is confusing. I wonder whether "trifluoroethyl" should be replaced by "trifluoromethyl", since the author has stated the importance of trifluoromethyl group in medicinal chemistry above this sentence.
2. Have the author ever detected the fluoride eliminated side product in the reaction process? The nickel-catalyzed defluorination might take place (*Chin. J. Org. Chem.* 2020, 40, 3307; *Org. Lett.* 2019, 21, 2723).
3. In the substrate scope Table 1, diverse nitroarenes were suitable for the four-component amination process. I wonder whether electron deficient anilines such as 4-trifluoromethyl/nitrile anilines could fit the reaction conditions. Either provide

deficient substrates or make a clear statement that they are not tolerated in this method.

4. In the substrate scope Table 3, the author used 2,3,3,3-tetrafluoroprop-1-ene (F4) to proceed the coupling reaction with moderate yield. Have the author ever tried 2-trifluoromethyl-1-alkene such as (3,3,3-trifluoroprop-1-en-2-yl)benzene? Also, I wonder whether the trifluoromethyl group could be replaced by other electron-withdrawing groups such as ester groups for example.

5. The author proposed that aryl aminal species S2 was likely to interact with low-valent Ni species (Ni⁰), resulting in the formation of a nickel(II) N-aryl-aminomethyl species (S4) in the mechanism study. To support this conjecture, the author cited reference 52 which developed a palladium-catalyzed vinylation reaction. This might be inappropriate.

6. According to the proposed mechanism, the coupling product and the essential intermediate S6 were synthesized through the radical-radical recombination. However, in general, the yield of radical-radical coupling is not high under photocatalytic conditions due to the relatively low concentration of radicals. The author should provide information on the stationary concentration of intermediate radicals to support this mechanism. Also, the radicals might be captured by low-valent Ni species and a nickel-catalyzed reductive elimination could not be ruled out.

7. Since the authors have demonstrated that there are up to eight radical active species in the system according to the proposed mechanism, the authors should provide evidence to prove whether the reaction is a nonchain/chain process. Moreover, the electrophilic/nucleophilic nature of the radical species should be discussed in the manuscript which might be helpful to explain the chemo-selectivity of the reaction.

Reviewer #3

(Remarks to the Author)

Fluorinated organic molecules have attracted great attention in pharmaceuticals and agrochemicals. Thus, the development of new catalytic strategies for the synthesis of fluorinated organic molecules has become an important topic in synthetic organic chemistry. In the present manuscript, the authors developed a methodology enabled by dual photoredox and nickel catalysis for the synthesis of N-trifluoroalkyl anilines through a four-component reaction. Employing easily accessible starting materials such as nitroarenes, tertiary alkyl amines, carboxylic acids in the radical-mediated coupling events and high levels of chemo and regioselectivity are the advantages of the protocol, while most of the isolated products are in the moderate to low yields. A vast variety of substrates (~ 113 examples) including pharmaceutical-relevant molecules and further derivatization of these products shows the potential applications. Based on their radical trapping experiment and other preliminary studies the authors suggested a feasible reaction mechanism, yet need to do more mechanistic investigations to get clear insights into the reaction mechanism. Also, the same authors reported previously three-component reaction using nitroarenes, tertiary alkyl amines, carboxylic acids for the synthesis of N-alkyl aniline products (ref. 46; Nat. Synth. 611 2, 1171–1183 (2023)). The present reported method is just an extension of their previous work and the generated alkyl radical from NHP esters can easily be trapped by electron-deficient alkene such as 3,3,3-trifluoropropene and further radical coupling events lead to the desired products. It's well demonstrated from the literature the addition of alkyl radicals to 3,3,3-trifluoropropene (Ref. 42; Nat. Commun. 13, 5539 (2022)). Accordingly, while this extension as described in the present manuscript does not reach the novelty standards required for publication in Nature Communication. I would like to suggest the authors need to recheck their statements before they submit the manuscript to other journals.

(1) In the Figure 4, e, they mentioned "C4-deuterium-enriched Hantzsch ester (HE-d2), it yielded the conventional product 14 without incorporating deuterium. The result implied that Radvi likely undergoes an inner-sphere electron transfer with low-valent nickel to form a Ni-(α -amino)trifluoroalkyl complex (S10). This complex is then protonated, releasing the target product." How the authors can avoid HAT from alkyl amine source?

is it possible to make the S7 through the alternative pathway and further utilize it to get more insights into the mechanism?

(2) Also, the authors mentioned in the manuscript quenching studies indicate the nitro arene quenching faster rate than Hantzsch ester, this result contradicts their previous studies published in Nat. Synth. 611 2, 1171–1183 (2023). Can they comment on these results, I would suggest to check the CV of the nitroarene (N14).

(3) the authors mentioned that "Notably, the transformation exhibited both chemo- and regioselectivity, without any detectable side-products from the two- or three-component reactions" Most of the isolated products are moderate to low yields can they comment on what is the mass-balance account for the observed low yields of the products.

Version 1:

Reviewer comments:

Reviewer #1

(Remarks to the Author)

The authors have addressed most of my concerns. But the citation in ref 47 is a preview of Shen's original research work: Reductive Quenching-Initiated Catalyst-Controlled Divergent Alkylation of α -CF₃-Olefin. Chem Catalysis 2022,2,1380-1393. The original research work should be cited. After that, I would like to suggest acceptance of this work by nature communications.

Reviewer #2

(Remarks to the Author)

The authors have expanded the discussion on reaction mechanisms in the revised manuscript which partly answered my questions. Also, electron-deficient substrates have been added as required. Considering the completeness of this article and its inspiration for synthetic organic chemistry, I recommend this work to be published on Nature Communications.

Reviewer #3

(Remarks to the Author)

The authors addressed most of the questions raised by the reviewers.

One of the reviewer requested to measure the reduction potential of the nitroarene (N14). Unfortunately the reviewer was unable to find the CV of the nitroarene (N14).

In their previous paper (Nat. Synth. 2023, 611 2, 1171–1183) the authors mentioned the reduction potential of the simple nitrobenzene $\{E_{1/2\text{red PhNO}_2/\text{PhNO}_2-\bullet}\} = -1.19$ V versus SCE in MeCN} and in the present paper the authors mentioned that reduction potential of electron-rich to electron-deficient nitro(hetero)arenes are approximately -0.8 to -0.6 V (vs SCE)

To avoid confusions, the reviewer strongly suggests to the authors to measure the reduction potential of nitroarene (N14) and add in the SI.

Before accepting this work, the authors need to clarify the above comment.

Topic: Response to the Referees

We sincerely thank the Reviewers for their time and effort in evaluating our paper. We have revised the manuscript in accordance with their valuable comments. Below is a detailed description of the changes made and our responses to the Reviewers' feedback.

Referee: 1

Recommendation: To be published after addressing the concerns.

Comment 1: Cheung, Ma and coworkers report a method for the synthesis of a diverse array of three-dimensional N-trifluoroalkyl aniline compounds via a metallaphotoredox-catalyzed multicomponent amination reaction that employs nitroarenes, tertiary alkylamines, 3,3,3-trifluoropropene, and carboxylic acids as the reaction partners. The substrate scope is broad. The capability for various derivatizations enhances both the structural complexity and functional diversity of the produced N-trifluoroalkyl anilines. In general, I like this chemistry, and I would like to suggest this work to be published in Nat. Commun. after addressing the following concerns:

Our response: We appreciate the Reviewer's positive feedback on our work.

Comment 2: 1) Recent synthetic application of trifluoromethyl alkenes should be cited: a) Chem Catal. 2022, 2, 1380-1393; b) Angew. Chem. Int. Ed. 2022, 61, e202212201; c) Angew. Chem. Int. Ed. 2023, 62, e202303218.

Our response: We have incorporated these relevant references as citations (47), (48), and (49) in the revised manuscript.

Comment 3: 2) How about the reaction of 1,2-disubstituted trifluoromethyl alkenes instead of 3,3,3-trifluoropropene?

Our response: We used 1,1,1,4,4,4-hexafluorobut-2-ene instead of 3,3,3-trifluoropropene in the reaction protocol. However, the desired four-component product was not formed (as shown below), likely due to the steric hindrance of the internal alkene, which impedes the addition of the alkyl radical.

Comment 4: 3) Radical clock experiment is suggested to be performed to support the proposed radicals.

Our response: We employed the redox-active ester (RAE) derived from hept-6-enoic acid as the reaction substrate for a radical clock experiment. The reaction afforded the ring-closing product, an *N*-trifluoroalkyl aniline featuring an incorporated cyclopentylmethyl group, in 25% yield. This outcome suggested the formation of a hex-5-enyl radical species, which rapidly cyclizes to produce the cyclopentylmethyl radical intermediate.

We have included the details of the radical clock experiment on page 11 of the revised manuscript:

“Additionally, the radical clock experiment employing the RAE derived from hept-6-enoic acid yielded the ring-closing product, an *N*-trifluoroalkyl aniline featuring an incorporated cyclopentylmethyl group. This outcome suggested the formation of a hex-5-enyl radical species, which rapidly cyclizes to produce the cyclopentylmethyl radical intermediate.”

We have included detailed reaction schematics and discussions in Fig. S1 of the Supplementary Information.

Comment 5: 4) “Macmillin and co-workers” should be “MacMillan and co-workers”.

Our response: We have corrected it to “MacMillan and co-workers” on page 2 of the revised manuscript.

Comment 6: 5) Supplementary Information: compound N31, 10, 29, 41, 42, 99, 107, ¹H NMR quantity is incorrect, please check carefully.

Our response: We have corrected the proton numbers in the ¹H NMR spectra for compounds N31, 10, 29, 41, 42, 99, and 107 in the Supplementary Information.

Comment 7: 6) Supplementary Information: compound R3, R24, 36 “HRMS (ESI) m/z: [M+H]+” should be “HRMS (ESI) m/z: [M+Na]+”.

Our response: We have corrected the notation to “HRMS (ESI) m/z: [M+Na]⁺” for compounds **R3**, **R24** and **36**.

Comment 8: 7) Supplementary Information: compound 95, “HRMS (ESI) m/z: [M+H]⁺ Calcd for C₂₅H₂₅F₃NO⁺” should be “HRMS (ESI) m/z: [M+H]⁺ Calcd for C₂₅H₃₅F₃NO⁺”.

Our response: We have corrected the notation to “HRMS (ESI) m/z: [M+H]⁺ Calcd for C₂₅H₃₅F₃NO⁺” for compound **95**.

Comment 9: 8) Supplementary Information. There are obvious impurities in compound **80** and **111** spectra.

Our response: We have purified compounds **80** and **111** and re-recorded the NMR spectra."

Referee: 2

Recommendation: Meet the bar for publication if the authors address the issues below.

Comment 1: This manuscript details a metallaphotocatalytic triple couplings for modular synthesis of elaborate N-trifluoroalkyl anilines. The yields of this transformation are generally good and the substrate scope is broad. Although this strategy has been applied to a three-component tandem reaction (Nat. Synth. 2023, 2, 1171), the authors achieved a more complex coupling reaction in this work, effectively avoiding the production of side-products from the two- or three-component reactions, which is still an improvement. Series of late-stage functionalization have been demonstrated, which are impressive and suggest that this process will likely be useful to the synthetic and medicinal chemistry community. In the mechanism study, a comprehensive suite of experiments was undertaken, which revealed the reaction intermediates and provided some evidence for the possible mechanism of these metallaphotocatalytic four-component amination process. However, due to the complexity of the reaction system, the study of the mechanism may need to be more detailed. Unquestionably, this is an important advance in the field of exploration of uncharted fluorinated, high C(sp³)-contented structural realms. Overall, I think this work will be of broad interest among the synthetic organic chemistry community, and meet the bar for publication on nature

communication if the authors address the issues below.

Our response: We appreciate the Reviewer's positive evaluation of our work. In response to the Reviewers' suggestions, we have expanded the discussion on reaction mechanisms in the revised manuscript. These additional experiments include the radical clock reaction, light/dark experiments, analysis of the defluorination side-reaction, and an evaluation of nickel catalysis and radical polarity in relation to chemoselective and productive bond formation, leading to product formation. We hope these results provide further insights into the reaction mechanism.

Comment 2: 1. In line 47, the author writes: "with the trifluoroethyl moiety being the most commonly incorporated group in drug design", which is confusing. I wonder whether "trifluoroethyl" should be replaced by "trifluoromethyl", since the author has stated the importance of trifluoromethyl group in medicinal chemistry above this sentence.

Our response: The term "trifluoroethyl" should be corrected to "2,2,2-trifluoroethyl" as it is more precise. The 2,2,2-trifluoroethyl group is a prevalent trifluoroalkyl scaffold in drug development, alongside the CF₃ group. We have replaced "trifluoroethyl" with "2,2,2-trifluoroethyl" on page 2 of the revised manuscript.

Comment 3: 2. Have the author ever detected the fluoride eliminated side product in the reaction process? The nickel-catalyzed defluorination might take place (Chin. J. Org. Chem. 2020, 40, 3307; Org. Lett. 2019, 21, 2723).

Our response: The following steps were taken to determine whether fluoride was generated from the reaction. An excess of saturated aqueous NaOH solution was added to the reaction mixture of the model reaction after the reaction was complete. The resulting mixture was diluted with excess ethyl acetate and stirred vigorously. This work-up was designed to neutralize the HF co-product formed and to allow for the formation of NaF in the event that fluoride is eliminated from 3,3,3-trifluoropropene. A small portion of the aqueous layer was extracted and dissolved in D₂O for ¹⁹F NMR spectroscopic analysis. A ¹⁹F NMR signal at -125.2 ppm was observed, indicating the likely formation of NaF, as compared to an authentic NaF sample (~ -122.6 ppm in D₂O, as shown below).

We hypothesized that some of the trifluoroalkyl radical (formed via the addition of the RAE-derived alkyl radical to 3,3,3-trifluoropropene) might undergo over-reduction, leading to fluoride elimination. As a result, 3 equivalents of RAE are required to increase the loading of the resulting trifluoroalkyl radical, thereby compensating for its

consumption via defluorination and, consequently, enhancing the product yield.

We have added the following discussion regarding the formation of the defluorination side-reaction on page 16 of the revised manuscript:

“Further ^{19}F NMR spectroscopic analysis suggested that fluoride is likely formed in the reaction (Supplementary Fig. 8). We surmised that the over-reduction of the trifluoroalkyl radical (**Rad-ii**) via photocatalysis or Ni catalysis is inevitable, leading to the generation of a trifluoroalkyl anion that results in fluorine elimination. However, the addition of excess RAEs and **F1** would ensure sufficient loading of **Rad-ii**, thereby compensating for the defluorination side-reaction and maintaining the reaction productivity.”

The experimental procedures and results have been added to Fig. S8 in the Supplementary Information.

^{19}F NMR spectrum (471 MHz, D_2O) of authentic NaF

Comment 4: 3. In the substrate scope Table 1, diverse nitroarenes were suitable for the four-component amination process. I wonder whether electron deficient anilines such as 4-trifluoromethyl/nitrile anilines could fit the reaction conditions. Either provide deficient substrates or make a clear statement that they are not tolerated in this method.

Our response: We utilized several electron-deficient nitroarenes in the reaction protocol. While nitrobenzene substituted with *p*-CF₃ afforded the product with a yield of ~20%, reactions with nitrobenzene substituted with *p*-CN, *p*-C(O)Me, and *p*-C(O)Ph did not yield any product. These highly electron-deficient nitroarenes were not tolerated in the reaction protocol and proved unproductive.

We have added the following content regarding the limitation of nitroarenes on page 5 of the revised manuscript:

“Highly electron-deficient nitroarenes, including 4-nitrobenzotrifluoride, 4-nitrobenzonitrile, 4-nitroacetophenone, and 4-nitrobenzophenone, were not tolerated and proved non-productive under the reaction conditions.”

Comment 5: 4. In the substrate scope Table 3, the author used 2,3,3,3-tetrafluoroprop-1-ene (F4) to proceed the coupling reaction with moderate yield. Have the author ever tried 2-trifluoromethyl-1-alkene such as (3,3,3-trifluoroprop-1-en-2-yl)benzene? Also, I wonder whether the trifluoromethyl group could be replaced by other electron-withdrawing groups such as ester groups for example.

Our response: We employed 3,3,3-trifluoroprop-1-en-2-yl)benzene as the fluorinated alkene in the reaction protocol, but no target product was formed. Similarly, the use of other 3,3,3-trifluoropropenes with 2-substituents such as Cl, Br, or CO₂Me did not yield the desired products, likely due to the steric bulk of these alkenes hindering the reaction. In addition, when the CF₃ group of 3,3,3-trifluoropropene was replaced with a CO₂Me group, the reaction proceeded, affording the corresponding product in ~10% yield. These results suggested that the CF₃ group is crucial for promoting both the efficiency and selectivity of the reaction.

Comment 6: 5. The author proposed that aryl aminal species S2 was likely to interact with low-valent Ni species (Ni⁰), resulting in the formation of a nickel(II) N-aryl-aminomethyl species (S4) in the mechanism study. To support this conjecture, the author cited reference 52 which developed a palladium-catalyzed vinylation reaction. This might be inappropriate.

Our response: To the best of our knowledge, the reaction of aminal with low-valent Ni(0) to form an aminomethyl-Ni(II) species is unprecedented. A similar reaction of

amine with Pd(0) to form an aminomethyl-Pd(II) species via oxidative addition has been documented in Ref 52 (Ref 55 now in the revised manuscript). Given the analogous reactivity patterns of Group 10 metal complexes, we speculated that the reaction of amine with more electron-rich Ni(0) via oxidative addition or nucleophilic substitution would be feasible. Therefore, this reference is cited to support the viability of this reaction pathway.

Comment 7: 6. According to the proposed mechanism, the coupling product and the essential intermediate S6 were synthesized through the radical-radical recombination. However, in general, the yield of radical-radical coupling is not high under photocatalytic conditions due to the relatively low concentration of radicals. The author should provide information on the stationary concentration of intermediate radicals to support this mechanism. Also, the radicals might be captured by low-valent Ni species and a nickel-catalyzed reductive elimination could not be ruled out.

Our response: We appreciate the Reviewer's suggestion to provide information on the stationary concentrations of intermediate radicals. Due to the lack of sophisticated instruments for direct concentration measurements, we employed TEMPO trapping experiments to evaluate the relative concentrations of the intermediate radicals. According to the HRMS analysis of the reaction mixtures, the adduct (T2) between TEMPO and the adamantyl-substituted trifluoroalkyl radical species **Rad-ii** exhibited the highest HRMS signal ($[M+H]^+ = 388.2829$), while other TEMPO-trapped species (T3–T7) did not show sufficiently strong signals. These results suggested that the adamantyl radical (**Rad-i**) and the adamantyl-substituted trifluoroalkyl radical (**Rad-ii**) are formed at much higher concentrations than other radical species.

Based on these findings, we proposed that Ni-mediated C–N and C–C bond formation reactions involving these radicals play a crucial role in the overall process, beyond the free radical interactions that contribute to product formation.

The revised discussion regarding the Ni-catalyzed bond formation reactions has been added on page 16 of the revised manuscript:

“Based on the HRMS analysis (Figs. 4a and 4b), the relative concentration of the trifluoroalkyl radical species S2 was significantly higher than that of the other detected radical species (**Rad-iii** to **Rad-vii**). This suggested that the concentrations of these other radical components are generally low. We proposed that low valent Ni ($Ni^{0/I}$)-mediated C–N and C–C bond formation reactions involving these radicals play a crucial role in the overall process, beyond the free radical interactions that contribute to product formation.”

Fig. 5 has also been revised to illustrate the involvement of low-valent Ni species ($Ni^{0/I}$) in mediating the C–N and C–C bond formation reactions.

Comment 8: 7. Since the authors have demonstrated that there are up to eight radical active species in the system according to the proposed mechanism, the authors should provide evidence to prove whether the reaction is a nonchain/chain process. Moreover, the electrophilic/nucleophilic nature of the radical species should be discussed in the manuscript which might be helpful to explain the chemo-selectivity of the reaction.

Our response: We conducted a light/dark experiment to assess the effect of light on product yield. In the presence of blue light, the product yields continued to increase, whereas in the absence of blue light, the yields remained steady (as shown below). This result suggested that the reaction occurs only during the irradiation period, implying that it likely proceeds via a non-chain photocatalytic sequence rather than a radical chain mechanism.

The trifluoroalkyl radicals (**Rad-ii'**) are electrophilic, and the *N*-aryl-aminomethyl radical (**Rad-v**) is nucleophilic. This polarity match between these two radical species would favor their coupling reaction, enabling chemoselective product formation.

We have added discussions on the chemoselective bond construction and the non-chain process on pages 15 and 16, respectively, in the revised manuscript:

“The electrophilic trifluoroalkyl radicals (**Rad-ii'**) and nucleophilic *N*-aryl-aminomethyl radical (**Rad-v**) efficiently couple to produce the *N*-trifluoroalkyl aniline products (**123**). The polarity match of these radical species underscores the chemoselective multiple bond connections that lead to product formation.”

“Moreover, the light/dark experiment demonstrated that the reaction occurs exclusively during the irradiation period (Supplementary Fig. 7), suggesting that it likely follows a non-chain photocatalytic mechanism rather than a radical chain process.”

The graphic result of the light/dark reaction has also been added in the Fig. S7 in the Supplementary Information.

Light/dark experiment

Referee: 3

Recommendation: The present manuscript does not reach the novelty standards required for publication in Nature Communication.

Comment 1: Fluorinated organic molecules have attracted great attention in pharmaceuticals and agrochemicals. Thus, the development of new catalytic strategies for the synthesis of fluorinated organic molecules has become an important topic in synthetic organic chemistry. In the present manuscript, the authors developed a methodology enabled by dual photoredox and nickel catalysis for the synthesis of N-trifluoroalkyl anilines through a four-component reaction. Employing easily accessible starting materials such as nitroarenes, tertiary alkyl amines, carboxylic acids in the radical-mediated coupling events and high levels of chemo and regioselectivity are the advantages of the protocol, while most of the isolated products are in the moderate to low yields. A vast variety of substrates (~ 113 examples) including pharmaceutical-relevant molecules and further derivatization of these products shows the potential applications. Based on their radical trapping experiment and other preliminary studies the authors suggested a feasible reaction mechanism, yet need to do more mechanistic investigations to get clear insights into the reaction mechanism. Also, the same authors reported previously three-component reaction using nitroarenes, tertiary alkyl amines, carboxylic acids for the synthesis of N-alkyl aniline products (ref. 46; Nat. Synth. 611 2, 1171-1183 (2023)). The present reported method is just an extension of their previous work and the generated alkyl radical from NHP esters can easily be trapped by electron-deficient alkene such as 3,3,3-trifluoropropene and further radical coupling events lead to the desired products. It's well demonstrated from the literature the addition of alkyl radicals to 3,3,3-trifluoropropene (Ref. 42; Nat. Commun. 13, 5539 (2022)). Accordingly, while this extension as described in the present manuscript does not reach the novelty standards required for publication in Nature Communication.

I would like to suggest the authors need to recheck their statements before they submit the manuscript to other journals.

Our response: We sincerely thank the Reviewer for thoughtful comments and suggestions. In response, we have conducted several additional experiments as recommended by the Reviewers to further explore the reaction mechanisms. These experiments included the radical clock reaction, light/dark experiments, analysis of the defluorination side-reaction, and an evaluation of nickel catalysis and radical polarity

in relation to chemoselective and productive bond formation, ultimately leading to product formation.

The development of this four-component reaction builds upon our prior work on a three-component reaction. This reaction has demonstrated high chemoselectivity, resulting in synthetically challenging and intricately structured *N*-trifluoroalkyl aniline compounds. Although the yields are modest to moderate due to the complexity of the multiple bond disconnection approach, these novel, complex, and lipophilic fluorinated compounds could serve as valuable synthetic scaffolds for the development of potent pharmaceuticals and agrochemicals.

We hope that these revisions, along with the potential utility of this reaction in the creation of new bioactive molecules, make the manuscript suitable for publication.

Comment 2: (1) In the Figure 4, e, they mentioned “C4-deuterium-enriched Hantzsch ester (HE-d₂), it yielded the conventional product 14 without incorporating deuterium. The result implied that Radvi likely undergoes an inner-sphere electron transfer with low-valent nickel to form a Ni-(α amino)trifluoroalkyl complex (S10). This complex is then protonated, releasing the target product.” How the authors can avoid HAT from alkyl amine source?

is it possible to make the S7 through the alternative pathway and further utilize it to get more insights into the mechanism?

Our response: To explore the possibility of hydrogen-atom transfer (HAT) from tertiary alkylamine to the α -carbon radical of the intermediate **Rad-vi**, we added 4 equivalents of triethylamine-*d*₁₅ to the model reaction (Fig. 4e, (ii) in the revised manuscript; Supplementary Fig. S6(b)). The product without D-incorporation was isolated, suggesting that HAT from tertiary alkylamines to **Rad-vi** for product formation is unlikely. This outcome is conceivable given that the α -C-H bonds of tertiary alkylamines are considerably strong (BDE for α -C-H of Et₃N: ~90 kcal mol⁻¹; BDE for α -C-H of Me₃N: ~93 kcal mol⁻¹), indicating that HAT from tertiary alkylamines could be both kinetically and thermodynamically unfavorable.

We have added a discussion on the reaction with triethylamine-*d*₁₅ on page 12 of the revised manuscript:

“In addition, the C-H bond of tertiary alkylamines could serve as another HAT reagent for **Rad-vi**. However, when excess triethylamine was added to the model reaction, no deuterium incorporation into product **14** was observed (Fig. 4e (ii)).”

We sincerely thank the Reviewer for the insightful suggestion to independently synthesize the imine species **S7** to examine its role as a reaction intermediate for product formation. The most appropriate approach to access imine **S7** would be a condensation reaction of aniline with the corresponding aldehyde bearing the trifluoroalkyl skeleton. However, this structurally complex aldehyde is not well-documented and would be synthetically challenging, thereby hindering further mechanistic study.

Comment 3: (2) Also, the authors mentioned in the manuscript quenching studies indicate the nitro arene quenching faster rate than Hantzsch ester, this result contradicts their previous studies published in *Nat. Synth.* 611 2, 1171–1183 (2023). Can they comment on these results, I would suggest to check the CV of the nitroarene (N14).

Our response: According to the literature, the reduction potentials of electron-rich to electron-deficient nitro(hetero)arenes are approximately -0.8 to -0.6 V (vs SCE; *Environ. Health Perspect.* **1985**, *64*, 309–320). The oxidation potential of the photoexcited iridium photocatalyst makes it sufficiently reducing $\{E_{1/2}^{\text{red}}[\text{Ir}^{\text{IV}}/\text{Ir}^{\text{III}*}] = -0.89$ V vs SCE $\}$, allowing for the thermodynamically favorable reduction of nitroarenes to initiate radical reactions. Based on this, we rationalized that the initial photoreduction of nitroarene is feasible.

Furthermore, the quenching experiments demonstrated that the photoexcited iridium photocatalyst preferentially reduces nitroarenes rather than oxidizes Hantzsch ester. This observation contrasts with our previous work, where the same photoexcited iridium photocatalyst preferentially oxidized Hantzsch ester rather than reduced nitroarenes. We believed this difference is likely due to the use of different reaction solvents in the quenching experiments. In this study, we used 1,2-dimethoxyethane/DMSO as the solvent, while 1,4-dioxane/*N*-methylpyrrolidine was used in the previous study. The variation in solvent polarity may influence the photocatalyst-substrate interaction, leading to different photocatalytic sequences in these two reactions.

Comment 4: (3) the authors mentioned that “Notably, the transformation exhibited both chemo- and regioselectivity, without any detectable side-products from the two- or three-component reactions” Most of the isolated products are moderate to low yields can they comment on what is the mass-balance account for the observed low yields of the products.

Our response: The model reaction using RAE based on 1-adamantanecarboxylic acid demonstrated selective formation of the target four-component product with a 76% yield, with no detectable side products observed by TLC analysis. However, the use of several tertiary and secondary RAEs, as well as all primary redox-active esters, generally resulted in modest product yields. This outcome was due to competition from two- and three-component reactions involving nitroarenes, tertiary alkylamines, and redox-active esters. The side products from several reactions were identified using ^1H NMR spectroscopy.

We surmised that the reaction's productivity is influenced by the nucleophilicity of the alkyl radicals. Tertiary alkyl radicals, being more electron-rich and nucleophilic due to their slightly inverted rather than planar structures, are more effective in promoting

reactions with electrophilic deficient 3,3,3-trifluoropropene and subsequent radical reactions. In contrast, primary alkyl radicals are less electron-rich and nucleophilic, which likely leads to impeded addition to 3,3,3-trifluoropropene but enhanced three- and two-component side reactions. Consequently, product yields derived from tertiary and secondary alkyl carboxylic acids are generally higher compared to those from primary carboxylic acids.

Furthermore, we speculated that the electronic effect of nitroarenes also plays a significant role in determining reaction yields. More electron-rich nitroarenes tend to react more productively, resulting in higher yields, while electron-deficient nitroarenes are presumably over-reduced, leading to the favorable formation of less reactive, highly reduced nitrogen-based intermediates, thereby hindering reaction productivity.

We have added a discussion on the effect of RAEs on product yield on page 7 of the revised manuscript:

“The use of several tertiary and secondary RAEs and all primary redox-active esters tended to result in modest product yields, due to competition from two- and three-component reactions involving nitroarenes, tertiary alkylamines, and redox-active esters.”

Topic: Response to the Referees

We sincerely thank the Reviewers for their time and effort in evaluating our paper. We have revised the manuscript in accordance with their valuable comments. Below is a detailed description of the changes made and our responses to the Reviewers' feedback.

Referee: 1

Recommendation: Suggest acceptance of this work by nature communications.

Comment 1: The authors have addressed most of my concerns. But the citation in ref 47 is a preview of Shen's original research work: Reductive Quenching-Initiated Catalyst-Controlled Divergent Alkylation of α -CF₃-Olefin. *Chem Catalysis* 2022,2,1380-1393. The original research work should be cited. After that, I would like to suggest acceptance of this work by nature communications.

Our response: We appreciate the Reviewer's acceptance of our work and sincerely apologize for the oversight. We have corrected reference 47 in the revised manuscript as follows:

Zhang, Y., Zhang, Y., Guo, Y., Liu, S. & Shen, X. Reductive quenching-initiated catalyst-controlled divergent alkylation of α -CF₃-olefins, *Chem Catal.* **2**, 1380–1393 (2022).

Referee: 2

Recommendation: Recommend this work to be published on Nature Communications.

Comment 1: The authors have expanded the discussion on reaction mechanisms in the revised manuscript which partly answered my questions. Also, electron-deficient substrates have been added as required. Considering the completeness of this article and its inspiration for synthetic organic chemistry, I recommend this work to be published on Nature Communications.

Our response: We appreciate the Reviewer's acceptance of our work.

Referee: 3

Recommendation: Clarify the comment before accepting this work.

Comment 1:

The authors addressed most of the questions raised by the reviewers. One of the reviewer requested to measure the reduction potential of the nitroarene (N14). Unfortunately the reviewer was unable to find the CV of the nitroarene (N14).

In their previous paper (Nat. Synth. 2023, 611 2, 1171–1183) the authors mentioned the reduction potential of the simple nitrobenzene { $E_{1/2}^{\text{red}} \text{PhNO}_2/\text{PhNO}_2\cdot^-$ } = -1.19 V versus SCE in MeCN} and in the present paper the authors mentioned that reduction potential of electron-rich to electron-deficient nitro(hetero)arenes are approximately -0.8 to -0.6 V (vs SCE)

To avoid confusions, the reviewer strongly suggests to the authors to measure the reduction potential of nitroarene (N14) and add in the SI. Before accepting this work, the authors need to clarify the above comment.

Our response: We appreciate the Reviewer’s positive feedback on our work and thank the Reviewer for the suggestion to perform cyclic voltammetry (CV) on 1-benzoxy-4-nitrobenzene (N14) to gain a clearer understanding of the photocatalytic reactions.

We have conducted the CV analysis of N14, determining the half-wave potential for its reduction to be -0.88 V vs SCE { $E_{1/2}^{\text{red}} [p\text{-BnOC}_6\text{H}_4\text{NO}_2/ p\text{-BnOC}_6\text{H}_4\text{NO}_2^-]$ = -0.88 V vs SCE in MeCN}, which lies within the redox window of the iridium photocatalyst. This finding further supports the viability of the proposed redox process. Additionally, the CV graph of N14 has been included in Supplementary Figure 9.

The following discussions on the reduction potential of N14 have been added to the revised manuscript:

“The half-wave potential for the reduction of N14 was determined to be -0.88 V vs SCE { $E_{1/2}^{\text{red}} [p\text{-BnOC}_6\text{H}_4\text{NO}_2/ p\text{-BnOC}_6\text{H}_4\text{NO}_2^-]$ = -0.88 V vs SCE in MeCN} (Supplementary Fig. 9), which falls within the redox window of the iridium photocatalyst, further supporting the viability of this preferential redox process.”

“This excited state can feasibly reduce nitroarene (N14) to radical anion species (N14’) { $E_{1/2}^{\text{red}} [p\text{-BnOC}_6\text{H}_4\text{NO}_2/ p\text{-BnOC}_6\text{H}_4\text{NO}_2^-]$ = -0.88 vs SCE} (Supplementary Fig. 9),⁶⁵ especially under conditions facilitated by HOTf through oxygen protonation.⁴⁶”